# Clinical Outcomes of Neoadjuvant Therapy in Human Epidermal Growth Factor Receptor 2 Breast Cancer Patients: A Single-Center Retrospective Study

**DOI:** 10.3390/jcm11051434

**Published:** 2022-03-05

**Authors:** Chih-Chiang Hung, I-Chen Tsai, Chiann-Yi Hsu, Hsin-Chen Lin

**Affiliations:** 1Division of Breast Surgery, Department of Surgery, Taichung Veterans General Hospital, Taichung 40705, Taiwan; mental7330@gmail.com (C.-C.H.); rjaeppfnh@gmail.com (I.-C.T.); 2Department of Applied Cosmetology, College of Human Science and Social Innovation, Hung Kuang University, Taichung 43302, Taiwan; 3Graduate Institute of Biomedical Sciences, College of Medicine, China Medical University, Taichung 40402, Taiwan; 4Biostatistics Task Force of Taichung Veterans General Hospital, Taichung 40705, Taiwan; chiann@vghtc.gov.tw; 5Division of Hematology and Oncology, Department of Medicine, Taichung Veterans General Hospital, Taichung 40705, Taiwan

**Keywords:** HER2 breast cancer, neoadjuvant chemotherapy and target therapy, pathologic complete response, overall survival

## Abstract

(1) Background: Neoadjuvant therapy is widely used to treat locally advanced breast cancer. It has been recently shown that it can also improve the prognosis of patients during the early stages of breast cancer. In the past, advanced breast cancer with positive Human Epidermal growth factor Receptor 2 (HER2+) resulted in poor prognoses; however, outcomes have since changed after the introduction of HER2-targeting therapy. Achieving pathological Complete Response (pCR) is the most important aim, as it is a predictor of long-term outcomes in high-risk breast cancer subtypes. (2) Methods: We performed a retrospective review of all breast cancer patients who were treated with neoadjuvant therapy at Taichung Veterans General Hospital (VGHTC) between 2010 and 2018. A total of 147 HER2+ breast cancer patients who underwent neoadjuvant chemotherapy involving anthracycline and taxane-based regimens were enrolled. Within that population, 95 and 52 cases received single-blockade (Trastuzumab) and dual-blockade (Trastuzumab and Pertuzumab) neoadjuvant anti-HER2 therapy, respectively. (3) Results: The dual-blockade therapy group displayed a significantly higher pCR rate after surgery as compared to the single-blockade group (63.5% vs. 43.2%, *p* = 0.019). Advanced stage, larger tumor size, lymph node involvement and HER2 expression status were associated with the pCR rate. The 4-year OS was 85.2% and 100% in the single-blockage and dual-blockade therapy groups, respectively (*p* = 0.041). (4) Conclusion: Anthracycline, followed by taxane-based neoadjuvant chemotherapy combined with the dual HER2-blockade, had a higher pCR rate and better outcome when compared with the single HER2-blockade strategy in locally advanced HER2 breast cancer.

## 1. Introduction

Breast cancer is one of the leading causes of death among females throughout the world, with a 5-year survival rate of merely 28% upon metastasis. Despite differences in genetic constitutions and lifestyles, Asian women and white women display similar breast cancer statistics [1]. For instance, the proportion of histology of breast cancer is similar, with the rate of acquiring cancer consistently rising until the age of 80 amongst women in both demographics. It is noteworthy that breast cancer is the most common cancer among women in Taiwan [2], with its mortality rate increasing proportionally with age. This poor disease prognosis created a scientific urgency to develop novel treatment methodologies. In the past, advanced breast cancer with positive Human Epidermal growth factor Receptor 2 (HER2+) resulted in a poor prognosis; however, outcomes have since changed after the introduction of HER2-targeting therapy. Outcomes also improved after incorporating target therapy with adjuvant and neoadjuvant chemotherapy in early and locally advanced HER2+ breast cancer patients [3,4]. Many prospective studies and even meta-analyses have confirmed the improved clinical results surrounding neoadjuvant HER2-targeting therapy [5,6]; however, real-world data in Taiwan is lacking. In this study, we report on the real-world outcomes of neoadjuvant therapy in locally advanced HER2+ breast cancer in Taiwanese female patients.

## 2. Materials and Methods

### 2.1. Patients

A total of 224 HER2+ breast cancer patients who underwent neoadjuvant chemotherapy at Taichung Veterans General Hospital (VGHTC) between 2010 and 2018 were recruited. All cancer specimens were examined by the pathologist at VGHTC. The neoadjuvant chemotherapy regimens were carried out following the surgeons’ preferences. Patients with tumors smaller than T1b were generally excluded from receiving therapy. In addition, patients were excluded if they failed to follow up with the treatment or if anthracycline-based chemotherapeutic agents were not given during the neoadjuvant therapy. This resulted in the enrollment of 147 patients for our retrospective cohort study. Of these 147 patients, 95 and 52 patients received single-blockade (Trastuzumab) and dual-blockade (Trastuzumab and Pertuzumab) neoadjuvant therapy, respectively. All patients were followed up on a regular basis at the hospital with their clinical data, including age, tumor stage, tumor size, lymph node involvement, Hormone Receptor (HR) status, HER2 expression status and pathologic diagnosis collected. Data regarding both Disease-free Survival (DFS) and Overall Survival (OS) were accumulated for analysis. DFS was defined as the period between diagnosis and recurrence or death, while OS was defined as the time between diagnosis and death. The patients’ data were collected from both the patients’ clinical records and from original pathology reports. These were obtained from the VGHTC research database which is managed by the Clinical Informatics Research & Development Center of VGHTC (Registered number CE19346A). It is important to note that the interpretations and conclusions contained herein do not represent those of VGHTC.

### 2.2. Neoadjuvant and Adjuvant Regimens

All neoadjuvant chemotherapy regimens used in this study were comprised of anthracycline, Pegylated-liposomal Doxorubicin (PLD) or Epirubicin. HER2-targeting therapy included Trastuzumab and Pertuzumab. All patients received four cycles of anthracycline (either Epirubicin or PLD) and Cyclophosphamide, followed by four cycles of Docetaxel accompanied with HER2-targeting therapy and vice versa. Each cycle comprised 21 days. The chemotherapy drugs were administered at the following dosages: PLD at 35 mg/m^2^, Epirubicin at 90 mg/m^2^, and Cyclophosphamide and Docetaxel at 600 and 75 mg/m^2^, respectively. The HER2-targeting therapy comprised Pertuzumab, which was administered at 840 mg in the first cycle and 420 mg in the following cycles. Finally, Trastuzumab was administered at 8 mg/kg in the first cycle and at 6 mg/kg in the following cycles. After their operations, all patients continued dual- or single-blockade therapy, the same as before the operations, for 1 year. For HR+ status, adjuvant hormone therapy was added with 10-year Tamoxifen for premenopausal patients and 5-year aromatase inhibitors for postmenopausal patients. For non-pathologic complete remission conditions, adjuvant chemotherapy was added according to the preferences of the surgeon.

### 2.3. Definition of Pathologic Complete Remission (pCR)

In this study, pCR was defined as the absence of any residual invasive carcinoma during the histological examination (Hematoxylin and Eosin) of the resected specimens, as well as in all regional lymph nodes examined after the completion of neoadjuvant systemic therapy (ypT0/Tis ypN0 in the current eighth edition of the AJCC Staging System).

### 2.4. Statistical Analysis

Baseline characteristics and intergroup comparison differences in categorical variables were analyzed using the Chi-square test or Fisher’s exact test. Univariate and multivariate statistical analyses were performed to study the association among pCR, age, clinical stage, tumor size, lymph node involvement, HR status, HER2 expression status, and pathologic diagnosis. DFS and OS were evaluated using the Kaplan–Meier estimator. The differences in survival status and recurrence amongst the groups were estimated using the log-rank test. The two-tailed t-test was used for statical analysis, with the level of significance set at 0.05. Datasets with a *p*-value < 0.05 were considered statistically significant. Statistical analyses were performed using the Statistical Package for the Social Sciences (IBM SPSS version 22.0; International Business Machines Corp, New York, NY, USA).

## 3. Results

### 3.1. Patient Characteristics

The clinical characteristics of the patients who were enrolled in this study are shown in Table 1. No statistical differences in age, clinical stage, tumor size, lymph node involvement, HR status, HER2 expression status, or pathologic difference were observed between patients in the single-blockade and dual-blockade groups.

### 3.2. The pCR Rate and Association with Patients’ Characteristics

The dual-blockade therapy group displayed a significantly higher pCR rate after surgery as compared to the single-blockade group (63.5% vs. 43.2%, *p* = 0.019, Figure 1). When compared to the pCR group, the non-pCR group displayed a higher clinical stage, larger tumor size, greater number of patients with lymph node involvement and HER2 expression with an Immunohistochemistry (IHC) score of 2+, which was further confirmed via fluorescence in situ hybridization (FISH+). The data are shown in Table 2. In our study, univariate analyses revealed that stage I vs. stage II (OR: 0.167; 95% Confidence Interval (CI): 0.036–0.781; *p* = 0.023) and stage I vs. stage III (OR: 0.068; 95% CI: 0.013–0.352; *p* = 0.001), tumor size ≤ 2 cm vs. >2 cm (OR: 0.307; 95% CI: 0.12–0.784; *p* = 0.014) and lymph node involvement “no” vs. “yes” (OR: 0.273; 95% CI: 0.126–0.591; *p* = 0.029) were all highly associated with pCR. Furthermore, multivariate analyses showed that in the HER2 expression status, IHC2+/FISH+ vs. IHC3+ (OR: 13.99; 95% CI: 1.378–142.004; *p* = 0.026) were highly associated with the pCR rate. Notably, although the pCR rate was higher in HR-negative patients than in HR-positive patients, it was not statistically significant (52.5% vs. 48.9%, *p* = 0.662) (Table 3). Interestingly, there were also cases changed from HER2 positive to negative after neoadjuvant therapy; however, the numbers were small. Only one case changed from HER2 IHC2+/FISH+ to HER2 IHC1+ in the dual-blockade group and only two cases changed in the single-blockade group (one from HER2 IHC2+/FISH+ to HER2 IHC1+ and the other kept HER2 IHC2+, while FISH was non-amplified).

### 3.3. Survival Analysis and Recurrence Differences between Single-Blockade and Dual-Blockade Therapy

The DFS in patients subjected to single-blockade or dual-blockade therapy is shown in Figure 2A. For the single-blockade therapy group, the median DFS was 79.2%, as compared to 84.3% in the dual-blockade therapy group, evidently highlighting a better trend of DFS in the latter. Nevertheless, the difference in DFS between the two groups was not statistically significant (*p* = 0.182). We then analyzed the OS amongst the patients in both single-blockade and dual-blockade therapy with the findings shown in Figure 2B. Additionally, no mortality was observed in 4 years of follow up in the dual-blockade therapy group. The 4-year OS was 85.2% and 100% in the single-blockade and dual-blockade therapy groups, respectively. Furthermore, the difference in OS between both groups was statistically significant (*p* = 0.041). We also found some differences of recurrent sites between the two groups. Although small in number, most patients who received dual-blockade therapy suffered from local recurrences. On the contrary, recurrence sites were evenly distributed in the local areas, lung, brain, bone and liver in the single-blockade group (Table 4).

### 3.4. Survival Analysis According to pCR

No instances of mortality were observed in the dual-blockade therapy group during the follow-up period which motivated us to assess the survival differences in the single-blockade therapy group according to pCR. The DFS amongst patients in single-blockade therapy is shown in Figure 3A. The 6-year DFS in non-pCR patients was maintained at 68.1% as compared to 90% in pCR patients. Furthermore, the difference in DFS between pCR and non-pCR patients was statistically significant (*p* = 0.008). The OS in the single-blockade therapy group according to pCR is shown in Figure 3B. The 6-year survival rate in non-pCR patients was 65.7% as compared to 85.7% in patients with pCR. The difference in OS between pCR and non-pCR patients in the single-blockade therapy group was statistically significant (*p* = 0.006). Though no mortality was observed in the dual-blockade therapy group, we analyzed the DFS in this group as per pCR. The 4-year DFS rate in non-pCR patients was 74% as compared to 91.4% in patients with pCR. The patients with pCR displayed a better trend in DFS; however, this difference was not statistically significant (*p* = 0.289).

## 4. Discussion

According to the GLOBOCAN 2020 statistics [7], there were 2.3 million new female breast cancer patients in 2020, which exceeded the number of lung cancer patients. Breast cancer patients make up 11.7% of all cancer patients. Quantitatively, breast cancer was responsible for 685,000 deaths, making it the fifth leading cause of cancer mortality worldwide. Based on the 2018 Cancer Registry Annual Report, breast cancer has been the most common type of malignant cancer and the leading cause of death in Taiwanese women. Although breast cancer has been viewed as a chronic disease, early detection and treatment have both helped improve the 5-year survival rate to 87% [8]. In order to prevent metastatic relapse in patients with non-metastatic breast cancer, a key surgical aim is to remove any tumor from the breast and lymph nodes. The treatment of non-metastatic breast cancer involves surgery and removal of the axillary lymph nodes, often accompanied by post-operative radiation. Systemic therapy comprises neoadjuvant (preoperative) and adjuvant (postoperative) therapies. These include (i) the use of endocrine therapy for treatment of most HR positive tumors, (ii) Trastuzumab-based HER2-directed antibody therapy in addition to chemotherapy for treatment of HER2+ tumors and (iii) sole chemotherapy for treatment of Triple-negative Breast Cancer (TNBC). Extended life and symptom palliation are the main therapeutic goals for patients with metastatic breast cancer. Additionally, surgery and radiation therapy are commonly used as a part of the palliative measures of systemic therapy in metastatic cases [9].

In locally advanced HER2+ breast cancer, neoadjuvant treatment is widely used because a positive response may indicate the need for less extensive surgery and thus improved surgical outcomes. After neoadjuvant treatment, pCR acts as a surrogate marker for predicting favorable outcomes, including Event-free Survival (EFS) and OS, according to large CTNeoBC pooled analysis [10]. In order to eliminate invasive cancer cells and to narrow the tumor lesions to achieve the goal of pCR, chemotherapy in neoadjuvant settings has been added to the novel therapeutic approach for treatment of invasive breast cancer. This is usually performed by administering anthracycline-based chemotherapeutic agents prior to surgery, which can achieve approximately a 50% pCR rate after full courses of neoadjuvant treatment are completed [5,11]. Pegylated-liposomal Doxorubicin (PLD), which is a characteristic of the liposomal-encapsulated formula, has improved the treatment by making it more precise. It also induces fewer adverse effects as compared to Doxorubicin, and for these reasons, it is commonly used in breast cancer therapeutics [12,13]. We have previously shown that when compared to Epirubicin-based neoadjuvant chemotherapy, PLD-based neoadjuvant chemotherapy offers similar treatment outcomes in addition to a reduced toxicity profile within the study’s follow-up duration [14].

The previous standards of care limited the use of chemotherapy regimens in neoadjuvant settings for HER2+ breast cancer treatment. However, recent developments in HER2-targeting therapy have greatly improved breast cancer treatment. For instance, the administration of Trastuzumab (a monoclonal antibody targeting the extracellular domain of HER2) to standard adjuvant chemotherapy has improved both the DFS and OS of HER2+ breast cancer patients. Furthermore, an increase in hazard ratios from 0.48 to 0.75 in randomized adjuvant trials favors the use of Trastuzumab-containing regimens [3,15]. An increasing number of studies have focused on the breakthrough effects of Trastuzumab in preventing cancer recurrences and in lowering mortality in HER2+ breast cancer patients. Furthermore, adjuvant Paclitaxel and Trastuzumab treatments have been associated with excellent long-term outcomes. Therefore, this combined therapy is now the standard of care for patients with small, node-negative HER2+ tumors [16]. The NOAH trial also confirmed the role of Trastuzumab in a neoadjuvant setting, as it significantly improved 3-year EFS compared to chemotherapy alone (hazard ratio: 0.59; 95% CI: 0.38 to 0.90) [17]. Therefore, Trastuzumab-based neoadjuvant treatment has since become the standard of care in locally advanced HER2+ breast cancer [18].

Pertuzumab has recently been further approved for both adjuvant and neoadjuvant treatment in HER2+ patients with early breast cancer after the approval of Trastuzumab. It is a recombinant humanized monoclonal antibody that binds to the extracellular dimerization domain II of HER2, which is located on the opposite side of domain IV, where Trastuzumab binds. By causing dual HER2 blockage, the combination of Pertuzumab and Trastuzumab has delivered beneficial results in patients with locally advanced cancer [19]. In the NeoSphere trial, administration of both Trastuzumab and Docetaxel resulted in pCR achievement in 29% of the women (95% CI: 20.6% to 38.5%). In contrast, 45% of women reported pCR achievement when administered with a combination of Pertuzumab, Trastuzumab and Docetaxel (95% CI: 36.1% to 55.7%; *p* = 0.0141). Furthermore, 24% of women administered Pertuzumab and Docetaxel achieved pCR (95% CI: 15.8% to 33.7%) in contrast to only 16.8% of women who reported pCR achievement when administered with both HER2-blockades without chemotherapy (95% CI: 10.3% to 25.3%) [20]. During the 5-year follow-up of those in the NeoSphere trial, there was no significant difference in DFS and Progression-free Survival (PFS) between the four groups, although patients who achieved total pathologic response had longer PFS (hazard ratio: 0.54; 95% CI: 0.29 to 1.00). The addition of Pertuzumab to Trastuzumab and Docetaxel improved pCR rates and provided better PFS, irrespective of total pathological complete response and HR status in subgroup analysis [21]. After dramatic results of Trastuzumab and Pertuzumab in HER2+ breast cancer, Pertuzumab was approved in Taiwan in 2013, followed by Trastuzumab approval 10 years later. However, Pertuzumab was only reimbursed in first line metastatic HER2+ breast cancer and restricted to combination with Docetaxel and Trastuzumab. Although Trastuzumab was reimbursed and wildly used in metastatic HER2+ breast cancer, it was only reimbursed for patients with lymph node involvement as peri-operative therapy.

Most studies and meta-analyses discussing the pCR rate reveal that incorporating anti-HER2 therapy in neoadjuvant regimens shows the pCR rate to be higher in HER2+ and HR status negative patients [6,10]; however, other risk factors, such as tumor size, status of lymph node involvement and clinical stage, which may or may not be related to pCR rate, are controversial. In the GeparQuattro study, although not statistically significant, smaller tumor size and patients without lymph node involvement at baseline had a trend toward higher pCR rate. [22]. In our study, we found that a higher clinical stage, large tumor size (more than 2 cm), lymph node involvement and HER2 expression status with HER2 2+/FISH+ are all risk factors associated with a lower pCR rate. In our cohort, the HR status negative patients showed a trend towards achieving a higher pCR rate; however, a statistically significant level was not met. This may be attributed to adding Pertuzumab into the neoadjuvant regimen, as there was a similar finding in the NeoSphere trial [20,21]. Regarding the explanation for a lower pCR rate in the HER2 2+/FISH+ patients in our study, a possible cause is that the patients with HER2 2+/FISH+ may have had a relatively lower HER2/Chromosome Enumeration Probe 17 (CEP17) ratio as well as different genetic heterogeneity [23]. Although HER2 2+/FISH+ patients were generally defined as HER2 positive, some studies have shown different clinical outcomes in this population. A retrospective study performed by Choi et al. [24] revealed that pCR was highly correlated with the HER2/CEP17 ratio. The median HER2/CEP17 ratio in pCR patients and non-pCR patients was 7.08 and 4.70, respectively (*p* = 0.03). Another large patient number retrospective study undergone by Kogawa et al. [25] revealed similar results. They found that as either a continuous variable or a cutoff level (more than 7), the HER2/CEP17 ratio could predict a higher pCR rate. Additionally, a higher HER2/CEP17 ratio could be an indicator of longer recurrence-free survival and OS. Our study had some similar results to those found in another Asian study. In a retrospective study performed in Japan, Takada et al. [26] also found that except for HR status, both clinical tumor size (T1-2 vs. T3-4, adjust odds ratio: 1.88, *p* = 0.002) and clinical node status (N0 vs. N2-3, adjust odds ratio: 0.65, *p* = 0.093) were predictors of pCR rate. In addition, patients who achieved pCR after Trastuzumab-based neoadjuvant therapy had longer DFS (*p* < 0.001).

Our study had some limitations. First, it was a retrospective study taking place in a single medical center with a relatively small number of patients, possibly resulting in some selection bias and a wide confidence interval. Second, we excluded patients undergoing non-anthracycline-based neoadjuvant regimens, such as TCH(P) (Paclitaxel/Docetaxel + Carboplatin + Trastuzumab with or without Pertuzumab), which is another standard neoadjuvant regimen. However, the number of patients who receive non-anthracycline-based regimens in our hospital is small due to surgeons’ preferences and this may increase the difficulty of analysis if we had not excluded those patients. Third, no mortality was noted in the dual-blockade therapy group during the follow-up period, so we cannot offer analysis on the relationship between survival and risk factors in patients who received dual-blockade. In addition, because of the reimbursement limitations, Pertuzumab was not given as recurrence therapy to most patients in the single-blockade group and different strategies were chosen after recurrence according to physicians, which may cause bias in the survival analysis. Further studies and longer follow-up times are still warranted, and we will be updating our results in the future.

## 5. Conclusions

In conclusion, this was a real-world study regarding neoadjuvant-targeting therapy in HER2+ breast cancer in Taiwanese women, focusing especially on the efficacy of neoadjuvant chemotherapy combined with dual-targeting therapy. We found that anthracycline followed by taxane-based neoadjuvant chemotherapy combined with dual HER2-blockade resulted in a higher pCR rate when compared with single HER2-blockade in locally advanced HER2 breast cancer patients who need neoadjuvant treatment. More advanced tumor features (clinical stages II and III, tumor size larger than 2cm, patients with lymph node involvement), as well as HER2 status expression with HER2 2+/FISH+ resulted in a lower pCR rate. Patients who received dual HER2 blockade and achieved pCR after neoadjuvant treatments had more favorable clinical outcomes for disease-free survival and overall survival.

## Figures and Tables

**Figure 1 jcm-11-01434-f001:**
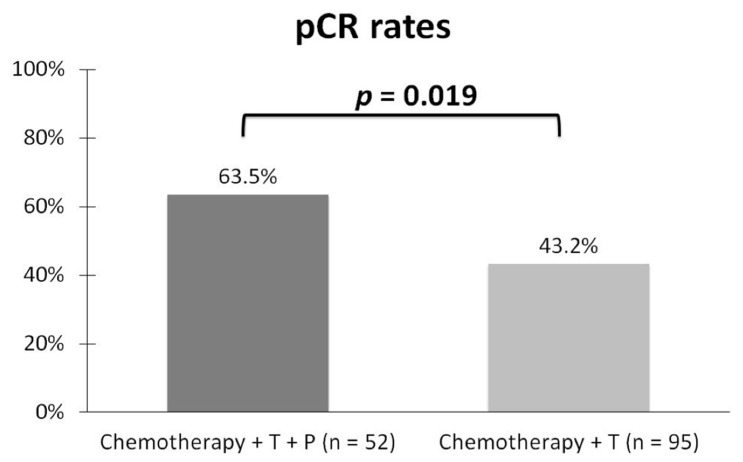
The significant differences of pathologic complete remission (pCR) after neoadjuvant chemotherapy combined with dual-blockade or single-blockade therapy. T, Trastuzumab; *p*, Pertuzumab.

**Figure 2 jcm-11-01434-f002:**
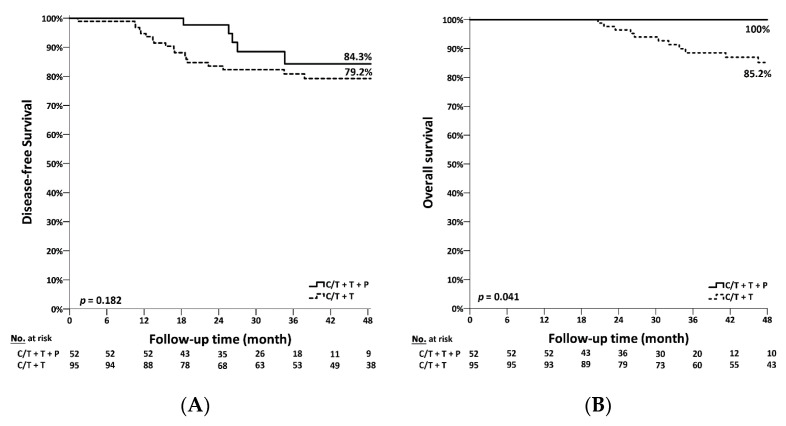
The survival curves of patients who received neoadjuvant chemotherapy with dual-blockade versus single-blockade therapy. (**A**) Disease-free survival. (**B**) Overall survival. C/T, chemotherapy; T, Trastuzumab; *p*, Pertuzumab.

**Figure 3 jcm-11-01434-f003:**
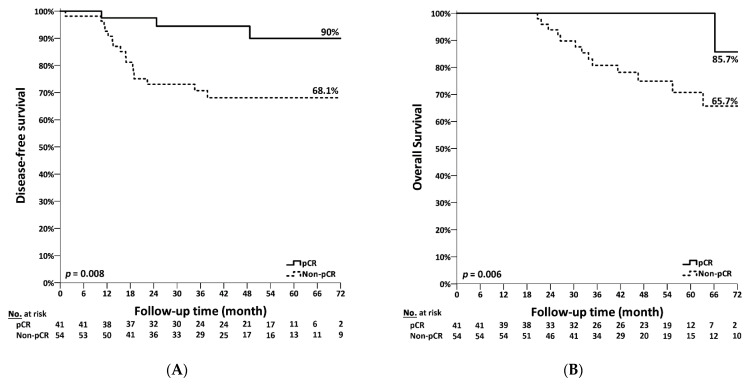
The survival curves of patients according to pathologic response after neoadjuvant chemotherapy, pCR (pathologic complete response) versus non-pCR. (**A**) Disease-free survival. (**B**) Overall survival.

**Table 1 jcm-11-01434-t001:** Baseline characteristics between single and dual target therapy.

	Trastumumab and Pertuzumab (*n* = 52)		Only Trastumumab (*n* = 95)		Total (*n* = 147)		*p* Value
Age (Diagnosis) (mean ± SD) t	50.58 ± 9.40		51.88 ± 9.72		51.42 ± 9.64		0.434
Age (*n*, %)							
<60	43	82.7%	74	77.9%	117	79.6%	0.490
≥60	9	17.3%	21	22.1%	30	20.4%	
Clinical stage (*n*, %)							
I	3	5.8%	12	12.6%	15	10.2%	0.361
II	37	71.2%	59	62.1%	96	65.3%	
III	12	23.1%	24	25.3%	36	24.5%	
Tumor size (*n*, %)							
≤2 cm	5	9.6%	21	22.1%	26	17.7%	0.071
>2 cm	47	90.4%	74	77.9%	121	82.3%	
Lymph node involvement (*n*, %)							
Yes	37	71.2%	67	70.5%	104	70.7%	0.936
No	15	28.8%	28	29.5%	43	29.3%	
HER2 expression (*n*, %)							
IHC2+/FISH +	6	11.5%	4	4.2%	10	6.8%	0.167
IHC3+	46	88.5%	91	95.8%	137	93.2%	
Hormone receptor status (*n*, %)							
Negative (ER− and PR−)	22	42.3%	37	38.9%	59	40.1%	0.691
Positive (ER+ or PR+)	30	57.7%	58	61.1%	88	59.9%	
Pathology (*n*, %)							
IDC	48	92.3%	92	96.8%	140	95.2%	0.327
ILC	2	3.8%	2	2.1%	4	2.7%	
Others	2	3.8%	1	1.1%	3	2.0%	

t T test. Chi-Square test; HER2, human epidermal growth factor receptor 2; IHC, immunohistochemistry; FISH, fluorescence in situ hybridization; ER, estrogen receptor; PR, progesterone receptor; IDC, invasive ductal carcinoma; ILC, invasive lobular carcinoma.

**Table 2 jcm-11-01434-t002:** pCR by characteristics.

	pCR (*n* = 74)		non-pCR (*n* = 73)		Total (*n* = 147)		*p* Value
Age (Diagnosis) (mean ± SD) t	50.65 ± 9.23		52.21 ± 10.05		51.42 ± 9.64		0.329
Age (*n*, %)							0.967
<60	59	79.7%	58	79.5%	117	79.6%	
≥60	15	20.3%	15	20.5%	30	20.4%	
Clinical stage (*n*, %)							0.001 **
I	13	17.6%	2	2.7%	15	10.2%	
II	50	67.6%	46	63.0%	96	65.3%	
III	11	14.9%	25	34.2%	36	24.5%	
Tumor size (*n*, %)							0.011 *
≤2 cm	19	25.7%	7	9.6%	26	17.7%	
>2 cm	55	74.3%	66	90.4%	121	82.3%	
Lymph node involvement (*n*, %)							0.001 **
Yes	43	58.1%	61	83.6%	104	70.7%	
No	31	41.9%	12	16.4%	43	29.3%	
HER2 expression (*n*, %)							0.009 *
IHC2+/FISH +	1	1.4%	9	12.3%	10	6.8%	
IHC3+	73	98.6%	64	87.7%	137	93.2%	
Hormone receptor status (*n*, %)							0.662
Negative (ER− and PR−)	31	41.9%	28	38.4%	59	40.1%	
Positive (ER+ or PR+)	43	58.1%	45	61.6%	88	59.9%	
Pathology (*n*, %)							1.000
IDC	70	94.6%	70	95.9%	140	95.2%	
ILC	2	2.7%	2	2.7%	4	2.7%	
others	2	2.7%	1	1.4%	3	2.0%	

t T test. Chi-Square test. * *p* < 0.05, ** *p* < 0.01. pCR, pathologic complete remission; HER2, human epidermal growth factor receptor 2; IHC, immunohistochemistry; FISH, fluorescence in situ hybridization; ER, estrogen receptor; PR, progesterone receptor; IDC, invasive ductal carcinoma; ILC, invasive lobular carcinoma.

**Table 3 jcm-11-01434-t003:** Univariate and multivariate analyses of pCR and patients’ characteristics.

		Univariate				Multivariate			
		OR	95% CI		*p* Value	OR	95% CI		*p* Value
Age	<60 vs. ≥60	0.983	0.441	2.193	0.967	1.051	0.419	2.637	0.915
Clinical Stage	I vs. II	0.167	0.036	0.781	0.023 *	0.374	0.039	3.601	0.395
	I vs. III	0.068	0.013	0.352	0.001 **	0.177	0.016	1.924	0.155
Tumor size	≤2 cm vs. >2 cm	0.307	0.12	0.784	0.014 *	0.639	0.170	2.402	0.507
Lymph node involvement	No vs. Yes	0.273	0.126	0.591	0.001 **	0.409	0.158	1.058	0.065
HER2 expression	IHC2+/FISH + vs. IHC3+	10.266	1.266	83.252	0.029 *	13.990	1.378	142.004	0.026 *
Hormone receptor status	Negative vs. Positive	0.863	0.446	1.67	0.662	0.844	0.404	1.763	0.651
Pathology	IDC vs. ILC	1.000	0.137	7.299	1.000	0.638	0.060	6.805	0.710
	IDC vs. other	2.000	0.177	22.564	0.575	2.610	0.21	31.706	0.451

Logistic regression. * *p* < 0.05, ** *p* < 0.01. pCR, pathologic complete remission; OR, odds ratio; CI, confidence interval; HER2, human epidermal growth factor receptor 2; IHC, immunohistochemistry; FISH, fluorescence in situ hybridization; IDC, invasive ductal carcinoma; ILC, invasive lobular carcinoma.

**Table 4 jcm-11-01434-t004:** Recurrence sites between single-target therapy and dual-target therapy. Some patients had more than one recurrent site.

	Trastuzumab and Pertuzumab		Only Trastuzumab	
	(*n* = 5)		(*n* = 19)	
Local (*n*, %)	4	80%	7	36.8%
Lung (*n*, %)	1	20%	5	26.3%
Brain (*n*, %)	1	20%	4	21.1%
Bone (*n*, %)	0	0%	4	21.1%
Liver (*n*, %)	0	0%	5	26.3%

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
