# Peer review of "Clinical Outcomes of Neoadjuvant Therapy in Human Epidermal Growth Factor Receptor 2 Breast Cancer Patients: A Single-Center Retrospective Study"

_jcm, 2022, doi:10.3390/jcm11051434_

Round 1

Reviewer 1 Report

  1. Conclusion needs to be more clear regarding authors recommendation.
  2. Some inappropriate associations were evaluated like HER2 2+ FISH positive vs. HER2 3+. Its not needed as both means HER2 positive.
  3. What do authors mean by early advanced cancer?
  4. What does tumor size means? is it postchemo size on pathological evaluation or prechemo clinical size. If its clinical than why an additional association was sought.
  5. Were there any cases of partial vs. no response.

Reviewer 2 Report

The authors performed a retrospective study of HER2 positive breast cancer patients treated with neoadjuvant chemotherapy (NAC). They compared the pCR rate and outcome after NAC between single-blockade (trastuzumab) and dual-blockade (trastuzumab and pertuzumab) neoadjuvant anti-HER2 therapy. They concluded that anthracycline, followed by taxane based NAC combined with the dual HER2-blockade, had a higher pCR rate and better outcome when compared with the single HER2-blockade strategy in early, locally advanced HER2 positive breast cancer. The subject and purpose of the study was clinically important, but there are some points that need to be discussed and clarified.

  1. The authors stated that the choice of treatment was based on the doctor's preference. Since the governmental approval for the use of trastuzumab and pertuzumab are different, could the physicians choose the appropriate treatment in the same situation?
  2. The authors should outline the Taiwanese guidelines for the use of trastumab and pertuzumab as an anti-HER2 therapy for advanced or recurrent breast cancer.
  3. The authors should include a description of the postoperative adjuvant treatment, especially for non-pCR cases.
  4. The authors should include a description of the duration of adjuvant endocrine therapy for HR positive cases.
  5. When discussing OS, the authors should describe the treatment method after recurrence. Did the authors use pertuzumab after recurrence in the trastuzumab alone group?
  6. The authors should explain the HER2 status of the tumors with non-pCR. Were there any cases where the HER2 status changed from positive to negative?
  7. The authors should describe the difference in the form of recurrence between trastuzumab alone and combination groups.
  8. An additional concern that the authors also pointed out is that the sample size is small. The authors stated that a chemotherapy (anthracycline followed by taxane) with dual HER2-blockade is effective in locally advanced cases, but this is inconsistent with the predominantly high pCR rate in N0 and small tumors.

Round 2

Reviewer 1 Report

Authors addressed the comments 

Reviewer 2 Report

The authors responded  appropriately to my comments.